# A Qualitative View of Homecare Support Workers on Unmet Health Needs of People with Dependency

**DOI:** 10.3390/ijerph17093166

**Published:** 2020-05-02

**Authors:** Jose Manuel Martínez-Linares, Francisco Antonio Andújar-Afán, Rocío Martínez-Yébenes, Olga María López-Entrambasaguas

**Affiliations:** Department of Nursing, Universidad de Jaén, 23071 Jaén, Spain; fandujar@ujaen.es (F.A.A.-A.); rmatinez@ujaen.es (R.M.-Y.); omlopez@ujaen.es (O.M.L.-E.)

**Keywords:** needs assessment, disabled persons, qualitative research, home nursing, homecare support workers, home health aides

## Abstract

Background: Longevity and population growth generate an increase in the number of people with dependency, who require homecare assistance to meet their health needs. Homecare support workers provide this care in Spain, and they may have unique insights into the unmet health needs of those receiving homecare assistance. The aim of this study was to determine the unmet health needs of people with dependency based on the perspective of homecare support workers. Methods: Qualitative exploratory-descriptive study. Through convenience sampling, homecare support workers from a Spanish province were selected, following inclusion and exclusion criteria. Four focus groups, transcription and thematic analyses were performed using Atlas.ti. Coding triangulation was carried out, applying criteria for scientific rigour. Results: The six themes obtained were classified into the material, psychoemotional, socioeconomical and psychosocial needs of people with dependency from the point of view of homecare support workers, along with the contributions of improvements and the need of these professionals for continuous training. Conclusions: People with dependency need complex technical assistance, materials, psychological attention due to their situation, and more effective assessments of their health and disability status. Homecare support workers perceive themselves to be essential in these assessments. They ask for psychological assistance, due to the emotional burden of their work, and believe this help would contribute to improving the quality of their service. Homecare support workers perceive that they are capable of performing their job, although they believe that some delegated activities are beyond their levels of competency.

## 1. Introduction

The European Council defined dependency as “the state of people who require assistance and/or important help to perform their activities of daily living, due to a lack or loss of physical, psychic or intellectual autonomy” [1]. The situations that lead people into such state of dependency are related to the increase in life expectancy, the decrease in mortality, the decrease in the birth rate and the increase in the rate of chronic diseases. All this allows an increasing number of people to reach an advanced age, with a higher risk of suffering from chronic diseases, which can cause complications that may create a permanent situation of dependency in the patient [2].

The degree of dependency of a person can be determined by assessment scales of activities of daily living (ADL), among which the Barthel’s [3,4] and Katz’s [5,6] indices stand out in our context (Andalusia, Spain), or by the assessment of instrumental activities of daily living (IADL), among which Lawton and Brody’s instrument is included [7]. A special and must-read report by the founders of the Home Alone Alliance^SM^ [8] about family caregivers concluded that the provision of care in these circumstances of dependency is full of complexities, diversity and intense care. Levine C., one of the co-authors of this report, points out the limitation that the classical scales have in terms of capturing the real range of tasks conducted by family caregivers of dependent people [9]. Pain, pressure ulcers, isolation, chronic illnesses, problems with medication use, limited activities of daily living and geriatric syndromes (e.g., malnutrition, depression, incontinence, delirium, sensory impairment, osteoporosis or fall-related injuries) [2,8,10,11,12,13] are some of the wide variety of problems faced by dependent people. The Spanish National Health Survey (2017) [14] shows results regarding the functional limitations of people over 64 years of age, with mobility and hearing impairment being the most prevalent limitations (45.32% and 44.10%, respectively) and the basic activities of daily living also being limited, with the most frequent limitation being in personal hygiene (16.96%), followed by in dressing (14.06%).

All these health-related conditions lead us to address human needs to be covered, that is, nursing. Jacqueline Fortin [15] explores the different conceptualizations that had existed of human needs since the first theorists’ proposals. She also approaches the human needs theories in nursing and specifically mentions three “needs theorists”: Virginia Henderson, Faye Glenn Abdellah and Dorothea Orem. However, we will not delve into this topic, since it is not the direct object of this study.

In 2002, the World Health Organization (WHO) provided a tool in order to standardise the description of health-related states and to measure the functioning, disability and health of an individual in a specific context. This WHO’s framework is the International Classification of Functioning, Disability and Health (IFN) [16], which focuses on classifying and diagnosing people’s level of health, and it is based on the biopsychosocial model of disability. Moreover, a detailed and practical checklist was created to record the information of an individual [17]. Spanish legislation related to dependency includes, in its Article 27, the IFN criteria to establish the degree of dependency of people [18].

The dependency rate in the Spanish population in the year 2019 was 54.28%, which had increased since 2008 [19]. By age group, the Spanish population over 64 years of age had also undergone a continuous increase from 2008, to 29.94% [20]. According to data from the Spanish Institute for the Elderly and Social Services (IMSERSO) [21], on 31 December 2018, a total of 1,767,186 people had applied for help for dependency, thus confirming the upward tendency in the number of applications. On 30 November 2019, a total of 1,111,559 people were receiving dependency benefits, of whom 54.22% were over 80 years of age and 82.20% were either severely dependent (grade II) or greatly dependent (grade III). The homecare assistance service attended to 326,043 users, that is, almost four out of 100 people of 65 years of age or older. On that date, day-care centers offered 90,577 places distributed across 3387 centers, of which 72,897 were occupied [22].

The homecare assistance service, as a social resource provided by local administrations, comprises “a series of preventive, formative and rehabilitation actions carried out by qualified people in the homes of the people in a situation of dependency, with the aim of attending to their basic activities of daily living” (this definition is originally in Spanish) [9]. The professionals who perform such actions are known as homecare support workers (HCSWs). They carry out services related to personal attention regarding ADLs (personal hygiene, eating, moving, going to the toilet, etc.), as well as services related to domestic needs (cleaning, laundry, cooking, etc.) [23]. In our context, this care is not provided by nurses. The professional training that HCSWs are required to have is included in the Official State Gazette (BOE) n° 317, through the resolution of 11 December 2017 [24] (as an example of the training content, Table 1 is attached). The labour regime of HCSWs is specifically regulated at the provincial level ([25], p.65). We will neither expand upon nor analyse the working situation beyond the commented generalities.

The home assistance of needs of a dependent person can be provided by family members, friends and/or homecare workers. However, this paper only focuses on the unexplored group of homecare workers.

People with dependency can have a series of unmet health needs, depending on the family member and social support they receive [27]. To date, such needs have not been explored from the perspective of HCSWs. The aim is to determine the unmet health needs of people with dependency, from a holistic view of the person, from the perspective of the HCSWs who assist them. Their opinions are interesting, since they spend time with them every day and help them with their ADLs and IADLs.

## 2. Materials and Methods

The methodology used in this study was based on the COREQ format [28] for qualitative studies.

### 2.1. Research Team and Reflexibity

The research team was composed of the following members, who had research training and previous experience in research projects: JMML (male), nurse and doctor; OMLE (female), nurse and doctor; RMY (female), nurse and master; and FAAA (male), psychologist and master. During the study period, JMML and OMLE worked as faculty members, RMY as a nurse and FAAA as a research technician. None of the research team members had any kind of relationship or previous contact with the participants of the study. The latter were informed about the composition of the research team and about the aim and interest of the study to show the unmet needs of people in a situation of dependency.

### 2.2. Research Context

This study was presented as a research project to a call for awards (see Acknowledgements). The project included the realisation of two different studies: one with HCSWs and the other one with dependent people. The Aging Lab Foundation funded this study and offered us the possibility to carry it out taking HCSWs as the informants, since they work attending to the dependent population in different localities of Andalusia (Southern Spain).

### 2.3. Study Design

Due to the complexity of the research objective and the difficulty of measuring the concepts, a qualitative descriptive-exploratory study design was selected [29], which allowed capturing the subjective perception of the participants [30]. This approach is based on the principles of naturalistic research, whose aim is to study individuals and phenomena in their natural state [31]. Using this approach, the authors minimise the interpretation and focus on describing the participant’s perspectives from their experience [32].

The participants, who were required to have at least one year of experience in this professional activity to be included in the study, were selected by convenience sampling.

A total of 30 people from the localities of Martos and Mancha Real (15 from Martos and 15 from Mancha Real), in Jaén (province of Andalusia, Spain), participated in the study. These two localities were selected for their proximity to the homes of the researchers.

Some people (women, as we did not find any men) rejected their participation offer stating that they did not have spare time for this research, whereas others claimed that they did not want to participate or that they did not have anything to contribute. It is worth mentioning that the vast majority of HCSWs in Spain are women, as stated in a national investigation report [26].

The data gathering was conducted in rooms of the social services community centres of both localities. Only the participants and the researchers were present during the gathering of data.

The main characteristics of the participants are detailed in Table 2. The common characteristics among all the interviewees are not provided in this table; these were sex and nationality (female and Spanish). The profile of HCSWs in Spain is in line with the one we obtained. The report that gathers this information [26] divides HCSWs into two groups according to years of experience: those with less than 10 years and those with 10 or more years. In both groups, the profile is that of a Spanish female (foreigners are a minority, except in Madrid and Barcelona), older women have completed primary education and those who have worked as a HCSW for less than 5 years have a different level of education.

Four focus groups were conducted (2 in Martos and 2 in Mancha Real) with which data saturation was reached [33]. The ad hoc script of questions was designed to obtain the information required to respond to the proposed objective, and it was created and revised by all the members of the research team. A pilot focus group was conducted with 3 HCSWs, after which no modification was required. Table 3 shows the main questions included in the script.

None of the four focus groups had to be repeated. They were all recorded in audio, and field notes were taken, which were incorporated in the data analysis. The duration of the focus groups was 1.5–2.5 h. The participants did not want to revise the transcriptions.

### 2.4. Data Analysis and Results

The coding of the transcriptions was conducted individually by two researchers, who then unified their encoded transcriptions to produce the final coding [34]. The content of the transcriptions was analysed following the method of the six phases of thematic analysis with scientific rigour described by Braun and Clarke (familiarisation with the data, the generation of initial categories or codes, ttheme search, theme revision, the definition and naming of themes, and the writing of the final report) [35], as well as the process to ensure the reliability of the results described by Nowell et al. [36]. The codes that were related to each other were grouped in categories, from which subthemes emerged, which, in turn, were grouped to produce the final themes, using Atlas.ti v.5 (Scientific Software Development GmbH, Berlin, Germany) for Windows©. The results were not sent to the participants for revision.

### 2.5. Ethical Considerations

The study was carried out following the ethical principles of the Declaration of Helsinki. The treatment of personal data was performed in compliance with Regulation 2016/679 of the European Parliament and the Council of 27 April 2016, on the protection of natural persons with respect to the treatment of personal data and the unrestricted movement of such data, which revokes Directive 95/46/CE.

This study was conducted after obtaining the approval from the Ethics Committee of the University of Jaén. Each participant was requested to sign the corresponding informed consent.

## 3. Results

The results are presented according to the six main themes that resulted from the analysis, which are summarized in Table 4.

### 3.1. Psychosocial and Emotional Sphere

From the perspective of the HCSWs, the suffering of many people with dependency is complex and involves many psychological and emotional aspects that have become a part of the attention they should provide.

#### 3.1.1. Loneliness as an Endemic Harm

The participants mentioned the loneliness that people with dependency feel, as an endemic harm that deteriorates their physical health state.


*“It is a pity; many of them live on their own and only see family members a couple of hours during the weekend or half an hour a day. Well, and these are the lucky ones! … One of my users’ daughters lives in another province and come to see her once a month”. (GF2PAAD-E2).*



*“There are clients who have someone to keep them company all day long, because they can afford that. Others just lock themselves in their homes and they only open the door to us; […] once, one of my patients called me on a Saturday morning requesting some bread, she run out of bread and had no one to go and buy it for her.” (GF3PAAD-E1).*


Therefore, they believe that many of these people demand greater attention to alleviate such feelings, which the HCSWs feel is partly their responsibility, although they state that they do not have enough time to provide it.


*“Some days I can’t even talk to them, simply because I have no time! And I believe it is essential to spend a moment with them.” (GF1PAAD-E2).*


The feeling of loneliness is expressed by the users to the HCSWs when they tell them that they would like them to spend more time with them. Likewise, this emotional reward also comes from the feeling of satisfaction generated from doing their best when working with these people.


*“We get bad moments, but we feel somehow rewarded when we see that we do our best in our job.” (GF2PAAD-E4).*


#### 3.1.2. Lack of Psychological Attention to People with Dependency

The participants stated that no psychological assistance is being currently provided to people with dependency to treat their situation of loneliness and symptoms like sadness or low self-esteem. It is highlighted that people only receive psychological help when there is an additional mental health issue or such situation impacts their emotional state in a serious way. The HCSWs stated that such attention is being replaced with drugs.


*“Emotionally, these people are in a very bad situation, and they tell themselves ‘I’m clumsy, I’m totally useless!’ I don’t know what to tell them to at least cheer them up” (GF2PAAD-E8).*



*“They give them antidepressants and things like that, and that’s it, to keep them calm so they are half-asleep all day long.” (GF1PAAD-E1).*


Likewise, the participants felt that the people with dependency and the relatives (and/or other careers) sometimes use their visits to vent their sorrows to them, since they all are sincere to the HCSWs about their physical and psychological state. However, neither dependent people nor their family members are completely honest with each other regarding their emotional status. HCSWs perceived themselves as the ones in the middle who really know the truth:


*“To a certain extent, they are sincere with us. Some people are alright with their families [some clients pretend that they are fine], but as soon as we arrive, they can’t take a single step [they complain about pain or another suffering], in many cases because they don’t want to disturb their families. In other cases, when we tell them that we will call their children to let them know what’s going on, they oppose to that.” (GF2PAAD-E3).*


#### 3.1.3. Role of the Family

The role that their relatives could play in the psychosocial wellness of these people seems to be essential.

##### Management of Bureaucratic Issues

The participants reported that the bureaucratic tasks and management of people with dependency are usually assumed and conducted by their relatives and that the users are happy with that.


*“Their children are the ones who manage the paperwork and all that.” (GF4PAAD-E5).*


##### Knowing the Real Needs

The acceptance of the situation of dependency by the families was also mentioned. The situation has a familiar impact that is not always managed in the most appropriate manner, especially in the case of people with Alzheimer’s disease. In many of these cases, the participants think that the relatives do not have a real perception of the needs of the dependent person.


*“In the case of patients with senile dementia or Alzheimer’s, I think that careers need guidance, because there are relatives who are more patient and mentally strong to endure and others not so much. Some even need anxiolytics, because the careers are with them every day!” (GF4PAAD-E8).*


Therefore, it is suggested that the involvement of the relatives in the attention and care of people with dependency may vary. Although most families take care, there are cases of families who are not really involved. The reasons for these cases were discussed during the focus groups—such as “the relatives do not have time to take care of him/her”, “the relatives just do not care” and “the relatives do not really know about the dependent person’s suffering and needs”—as they tend to hide their needs from their relatives to avoid the feeling of being a burden to them.


*“Many of the family members do not really know what their father or their mother needs. Sometimes they just do not want to know it, and we try to tell them…but we are generally ignored, and the patients do not help because they usually hide their real pain or sadness in front of them, but then they cry when their children are not there!” (GF1PAAD-E5).*



*“They say that they do not want to be a “burden” to their children; they say that their children are too busy with work and attending their own children” (GF2PAAD-E6).*


#### 3.1.4. Fear of Hospitalisation

According to the HCSWs, people with dependency and their relatives and/or informal careers fear hospitalisation, due to the subsequent deterioration of their mood and physical health; this would also undo the progress achieved at home. Therefore, they do all they can to prevent it.


*“Their relatives tremble, because taking them to the hospital makes them think ‘My goodness, how will he/she come back?’ It’s always the same conversation. The children have to take turns, they don’t know how long they will stay; they come back home but with worse health, less strength, worse mood, etc.” (GF3PAAD-E7).*


### 3.2. Legislation and Financial Issues

HCSWs were asked whether they thought the users had any problems with financial resources and, if so, how they noticed that impacted their health needs.

#### 3.2.1. Importance of the Dependency Law

The recognition of dependent persons through the mechanism of the Dependency Law has a positive impact on the satisfaction of some of their needs through the provision of services rather than economic benefits.

However, according to the HCSWs, not all the people with dependency have received the necessary recognition as such and the benefits provided by the Dependency Law are not always assigned correctly.


*“There are cases in which the benefit is granted because the dependent person has been in the waiting list for many years, but in other cases they give it to someone who is fine, someone who goes to the bar to have a glass of wine, goes shopping... And then, we see some people who can’t move from their beds, who eventually die without receiving anything at all…” (GF1PAAD-E4).*


Similarly, the participants mentioned that the benefits provided by the Dependency Law take too long to reach the people who are recognised as dependent, with some of these people being in a state that does not allow such delay, especially in the case of benefits to make adjustments in the home of the dependent person. The HCSWs stated that, during the last economic crisis, there was a decrease in the granting of benefits, as well as a greater delay in their provision.


*“I know several cases in which the benefit arrived when the person had already died. It depends on the assistance required. Some years ago it was worse... They only provided the basics!” (GF3PAAD-E6).*


#### 3.2.2. Greater Expense with Greater Dependency

In line with the Dependency Law comments, from the perspective of the interviewees, people with a high degree of dependency have unmet basic needs related to the assistance material they require. This lack of resources makes it hard to meet the need of users to move and can also diminish their recreational options.


*“Some heavy people may have an adjustable bed, but still lack an electric crane, so it is not possible to take them to the living room to watch TV, for example”. (GF2PAAD-E3).*


However, in some cases, the architecture of their homes requires adaptations in order for these devices to be functional, which further exacerbates the situation. Thus, the material needs are met based on the complexity and cost of the material. The participants stated that more material was provided before the budget cuts derived from the economic crisis.


*“Since it is a great expense for the Social Security, if they grant you the crane, then they don’t give you the bed. But previously [before the economic crisis] they used to provide both.” (GF1PAAD-E4).*


The case of people having a low–moderate degree of dependency is different. They have handles or grabs in their bathtubs to facilitate their personal hygiene and walking frames to assist their movement with a reduced risk of falling. These kinds of material are always provided, as stated by the participants: “because these are not expensive”. Similarly, they all have the so-called “button” of the Telecare Service for emergency situations.

Furthermore, there are cases in which people with dependency do not make use of the assistance material that was provided to them. The participants stated that there are a minority of people who have more assistance devices than they need and, consequently, such material is not being used correctly.


*“For example, I attend to a young woman who needs practically everything. And she has just what she needs. But there are things that she does not use, such as the crane […] She has a crane and a chair, but she only uses the crane to hang the towels.” (GF2PAAD-E6).*


Lastly, there are people with dependency who, due to their economic situation, cannot satisfy other needs, which can be related to personal hygiene products, clothing, medicines or specific assistance materials.


*“Some families go basic with everything: creams, clothing... There is a woman who has two sets of clothes; I have to put one to dry in the radiator while she wears the other, because she doesn’t have any more clothes…” (GF3PAAD-E2).*



*“In some cases, they have to pay half the price of their medicines, especially when they are very expensive, and not just that; if they need some kind of rehabilitation or diapers, they have to pay it themselves fully.” (GF2PAAD-E6).*



*“They frequently need to buy things, such as anti-decubitus mattresses or anti-decubitus cushions..., and that’s expensive.” (GF4PAAD-E8).*


### 3.3. Possible Solutions Suggested by the HCSWs

The participants provided a series of solutions to contribute to satisfying the needs of people with dependency.

#### 3.3.1. No Benefit Increase

According to the HCSWs, a possible solution for the unmet needs of people with dependency would not be related to an increase in the benefits they receive, but to an increase in the availability of resources and a faster provision of these.


*“If they are offered more money, they would just take it. As HCSW, we believe that receiving more material resources, or getting more hours of professional assistance, would be a lot more helpful than simply getting more money.” (GF2PAAD-E5).*


#### 3.3.2. Involvement of HCSW in the Assessments and Re-Assessments

Since the HCSWs are the ones who spend more hours in direct contact with dependent people, who they have to attend to continuously, the participants stated that they would like to be taken into account in the assessment or re-assessment of the type of assistance that each person requires, based on their state and degree of dependency. They are not considered and, thus, feel undervalued. Similarly, they believe it is necessary to carry out more frequent re-assessments of the needs of people with dependencies that require assistance, involving the HCSW.


*“I have seen people in their homes who do a great acting performance in the presence of the social worker! And if the social worker decides that the person has certain needs, that’s what counts... But if the HCSW says that that’s not the usual situation, it doesn’t matter. Our opinion is not taken into account.” (GF1PAAD-E7).*



*“A social worker can write a report with 15 min of observation? That’s not enough time to establish the needs of that person. It’s impossible.” (GF1PAAD-E3).*


### 3.4. In Need of Continuous Training

The researchers asked questions about the perception of HCSWs of their capacity to detect and attend to risk situations, or life-threatening conditions, in people with dependency.

#### 3.4.1. Capacity to Recognise Versus Capacity to Act

The HCSWs—who had different levels of official regulated education, although all of them had primary education—perceived that their level of training and capacity was adequate to attend to people with dependency and to recognise emergency situations that may occur, since they attend to them every day and can detect health problems that may appear suddenly (hypoglycemia, hypotension, dyspnea, syncope..., etc.). Moreover, they claimed to know how to act in such an emergency situation and highlighted the importance of the telecare “button” in these cases.


*“The user and I fell down the stairs together once... Thank goodness she had the button here [points at her chest]! She landed on me, and I couldn’t move any of us two. She pressed the button and the ambulance came…” (GF2PAAD-P2).*



*“They can choke while being fed! […] And we have to get them to cough hard, while hitting them lightly in the back…” (GF3PAAD-E5).*


#### 3.4.2. Performing Delegated Activities

The nurses of primary healthcare delegate to HCSWs activities for which the latter have not been trained, especially in the case of wound cleansing and dressing or managing the pressure ulcers that people with dependency may have. The participants claimed that they do not feel they have enough training to perform these medical tasks.


*“Of course, if something goes wrong, it’s our fault! If that person needs someone to come every day to treat their wounds, they can’t expect us to do that. No!” (GF2PAAD-E7).*


#### 3.4.3. Asking for More Knowledge

Although these professionals receive continuous training courses and most of them are satisfied, some believe that these are not enough and that they need to receive more, with contents related to mobilisation and the prevention of lesions, the fundamentals of nutrition, basic knowledge of the main diseases derived from disability, and cardiopulmonary resuscitation.


*“It’s good to remember, and we need more courses. We need to learn more and improve. It would be great if there were more courses available.” (GF1PAAD-E3).*


## 4. Discussion

All the European models [37] for attention to dependency have the same type of benefits provided to people who receive such recognition: economic aid, places in hospices and day-care centres, and homecare assistance services provided by HCSWs. The aim of this study was to understand the unmet health needs of people with dependency from the perspective of the HCSWs who attend to them in their homes.

The perception of the participants of the present study of the unmet needs of people with dependency was mainly related to the psychological/emotional sphere, and about resourcefulness in the case of those with a high degree of dependency. They did not mention any unmet biological or physiological needs in the people they attend to; however, they did mention the fears that these patients expressed when they thought about hospitalisation. In fact, the possible negative consequences of being hospitalised have been reported in older people [38].

All the HCSWs highlighted throughout the interviews the feeling of loneliness of dependent people. This is in line with the data provided since years ago, by studies carried out in Spain and in other European countries [39]. Back then, the number of people who claimed to feel lonely ranged between 5% and 6% in Denmark and Sweden, and between 10% and 14% in Great Britain, Belgium, France, Ireland, Luxemburg and Spain. These percentages have increased in more recent comparative studies [40]. Therefore, the loneliness that dependent people feel can be considered as an endemic harm [41]. In Spain, there are currently 4,732,400 people living alone, of whom 2,037,700 (43.1%) are 65 years of age or older. Of these, 1,465,600 (71.9%) are women [42].

The effects of unwanted loneliness have been demonstrated. Among the most recent results, it has been reported to affect the health and the quality of life of elderly people, since it is associated with a worse perception of one’s own health, the existence of physical limitations, and multimorbidity [43]. Depressive disorders and anxiety have also been related to loneliness and low social support [44,45,46]. Moreover, the risk of death can increase by 26% [47].

Care-giving for dependent elderly people requires collaboration between HCSWs and family caregivers [48]. This issue has not been particularly explored in this research, although the interviewees expressed their opinion regarding the role that informal family carers played in the satisfaction of the needs of their dependent relatives. While bureaucratic tasks related to financial or medical issues are covered by the family (normally the children), HCSWs think that their social and/or emotional needs are generally not covered. “They do not want to be a burden” is what most interviewees said, and “they do not tell the family what they really need” is one of the reasons attributed to the unmet psychoemotional needs. The burden of care in elderly people has been widely documented, with a special focus on the burden felt by family caregivers [48,49,50].

Limitations in the availability of complex technical assistance materials have been found. People with a high degree of dependency require adjustable beds and cranes, or even home adjustments, that would improve their mobility [25] and, as stated by some of the participants, might improve the recreational options of these people. The lack of adaptation of the environment to the mobility needs of people with dependency was already reported by Grande and González [51]. According to these authors, the assistance material that these people had was mainly basic: walking sticks (23%), walking frames (18%), crutches (16%) and wheelchairs (10%); these results are in line with the opinions of the participants of our study when they said that low–moderately dependent people have their material needs covered.

The HCSWs also pointed out that people with dependency have additional expenses due to their situation, and that such expenses increase with the degree of dependency, due to the technical support required, the home adaptations needed or the hiring of careers, mostly. The estimation of the economic cost of informal care services provided in Spanish homes in the year 2008 ranged between 23.06 billion and 50.16 billion euros, which would have represented 2.1% of the gross domestic product of that year [52].

Receiving attention from HCSWs does not guarantee that all the needs of these people will be covered. This situation was mentioned in the study conducted by Franco and Ruiz [25], who analysed the needs perceived by dependent people in different Spanish cities. The lack of assistance hours in the schedules of HCSWs is usually covered by the relatives of the persons with dependency, or with the private hiring of carers (in most cases with no training), which has also been reported in another Spanish study [53]. In the present study, the participants stated that much of this time, which is not covered by them, is covered by no one and that the patients are lonely.

The lack of time mentioned by the HCSWs forces them to prioritise the activities related to the basic physiological needs of the person and to take care of the house, leaving no time for psychoemotional attention. However, both types of attention are considered to be the responsibility of HCSWs in the training courses that these professionals undertake in their education [54]. However, the Dependency Law [18] specifies that the actions provided by the homecare service for people in a situation of dependency are services related to personal attention in the realization of ADLs and services related to attention to domestic needs.

We are now in 2020, and in 2002, Tomás et al. [55] had already come to the conclusion that “the model of care for the elderly in Spain should develop emotional support services, economic and social benefits to the family and to pay special attention to dependent people who live alone and with scarce material resources”.

A study conducted in Spain [53] reported that HCSWs require continuous training to improve their professional performance, especially in this homecare activity in which they must carry out general tasks related to basic needs (eating, dressing, personal hygiene, etc.), as well as more complex duties (auxiliary nursing care, psychological support, technical tasks, etc.). However, the study carried out by the Spanish Ministry of Employment, Migration and Social Security [56] indicated that the training offer is targeted to professionals with contracts of over 1 year in length and that the reception of this offer is unequal, due to the existence of workers who do not have a “culture of continuous training” along with others who demand it and even take courses on their own, which they pay for with their own money. According to another Spanish study [57], the training to access this type of job does not match the offer of continuous training for it, and this depends on the public administration that regulates such services and on the size of the city hall. It is also stated that, in general terms, both initial access training and update training are offered by 51% of the Spanish city halls.

Investigations conducted with nurses in the USA [58] and in Ireland [59] detected elderly self-neglect among their clients. With regard to self-care, nursing has its reference in Dorothea Orem’s theory [60], which defines it as the set of practices performed by a person on his/her own will to preserve his/her well-being, health and life. Hale and Marshall [61] gather the aspects related to healthy aging published by U.S. Department of Health and Human Services: “staying active, ensuring a safe home environment, engaging with others, eating healthy, staying well hydrated, maintaining mental health, and managing medication and treatments”. None of these aspects of the general care of people, which would be framed within the nursing competencies [62], were mentioned by the participants of this study, although they do not have such training either. This statement is not judgmental, as it merely aims to reflect the importance of having interviewed HCSWs, who take care of the well-being of dependent people and have their own unique perceptions.

In view of the results of the present study, actions should be implemented to improve the homecare assistance provided to people with dependency.

### Limitations

This paper focuses on the perception of HCSWs about what they consider the unmet needs of dependent people. However, such perception has been explored in nurses’ perspectives in different countries and may have substantial differences, as nurses are specialists in approaching and attending to human needs, unlike HCSWs.

The authors would have liked to interview some male HCSWs, but none were available.

## 5. Conclusions

From the perspective of HCSWs, the needs of people with dependency are also related to complex technical assistance materials, psychological attention due to the situation of dependency and the feeling of loneliness, the assessment of their degree of dependency and the reception of the benefits and support that are assigned to them. The involvement of HCSWs in the assessment and re-assessment for the recognition of the level of dependency, as well as them receiving psychological attention to tackle the emotional stress they endure in their job, would improve the quality of the service they provide.

HCSWs perceive that they have the capacity to detect and prevent situations of risk and emergency in the people with dependency they attend to. However, they do not believe they are qualified to perform activities delegated by other professionals and that do not fall within their competence. Therefore, with the aim of increasing their knowledge on the basic homecare of people with dependency, they ask for continuous training.

The authors of this paper would like to share with the public a reflective question: What if nurses were involved in homecare support in Spain? Would it improve the quality of care of dependent people?

## Figures and Tables

**Table 1 ijerph-17-03166-t001:** Training that qualifies a person to work as a homecare support worker (HCSW).

Certificate	Training Modules	Duration
Certificate of Professionalism in Home Social Health Care for People	Home hygiene and health care	170 h
	Home psychosocial attention and support	210 h
	Home attention and family nutrition	100 h
	Non-labour practices	120 h

Source: prepared by the author from [26].

**Table 2 ijerph-17-03166-t002:** Sociodemographic characteristics of HCSWs.

Code	Age	Civil State	Education Level *	Experience Years	Number of People Attended to **
GF1PAAD-P1	49	Married	Higher	7	13
GF1PAAD-P2	54	Married	Secondary	10	14
GF1PAAD-P3	53	Married	Secondary	11	8
GF1PAAD-P4	47	Married	Secondary	2	10
GF1PAAD-P5	49	Divorced	Secondary	10	12
GF1PAAD-P6	34	Single	Secondary	12	9
GF1PAAD-P7	26	Single	Secondary	2	12
GF1PAAD-P8	44	Married	Secondary	3	7
GF2PAAD-P1	55	Married	Secondary	10	18
GF2PAAD-P2	50	Married	Secondary	5	10
GF2PAAD-P3	32	Single	Secondary	11	10
GF2PAAD-P4	53	Married	Primary	15	6
GF2PAAD-P5	54	Married	Primary	13	6
GF2PAAD-P6	51	Married	Primary	7	14
GF2PAAD-P7	38	Married	Primary	7	15
GF3PAAD-P1	37	Married	Secondary	5	12
GF3PAAD-P2	44	Single	Secondary	12	12
GF3PAAD-P3	44	Married	Primary	1	11
GF3PAAD-P4	39	Married	Secondary	3	8
GF3PAAD-P5	48	Married	Primary	5	7
GF3PAAD-P6	50	Married	Secondary	11	8
GF3PAAD-P7	46	Married	Primary	4	7
GF4PAAD-P1	43	Married	Secondary	8	10
GF4PAAD-P2	48	Divorced	Primary	10	7
GF4PAAD-P3	33	Married	Secondary	10	8
GF4PAAD-P4	49	Married	Primary	7	8
GF4PAAD-P5	40	Divorced	Primary	8	9
GF4PAAD-P6	30	Single	Secondary	12	7
GF4PAAD-P7	35	Single	Primary	8	9
GF4PAAD-P8	40	Married	Primary	17	8

Abbreviation: HCSWs, homecare support workers. * Level of official regulated education, excluding continuous training. ** Number of people with dependency they attend to in their homes weekly. Source: developed by author.

**Table 3 ijerph-17-03166-t003:** Main questions asked in the focus groups.

Pre-established Categories	Questions for HCSW
Basic needs	Do you think the basic needs of people with dependency are met (eating, dressing, moving, going to the toilet...)? What help do they require for these needs to be satisfied?
Availability of home assistance material	Do the chronic patients you attend have any sort of sanitary and/or assistance material in their home due to their chronic disease (adjustable bed, walking frame, wheelchair, etc.)? Do you think that the material they have is enough or do they require further material? If so, which material?
Bureaucratic management and processes	How frequently must chronic patients take care of paperwork related to their chronic disease? If they require assistance to do this, do they usually have someone who can help them?
Personal and family economy	Do you perceive that the economic situation of the chronic patients you attend has changed due to their chronic disease? In your opinion, what caused such change in their economic situation? Do you think their chronic disease might have influenced this?
Emotional attention to people with dependency and their relatives	How do you think that chronic disease has emotionally and socially influenced the daily life of the people you attend? Do they have a career at home? How do you think that their chronic disease has emotionally and socially influenced the daily life of their careers?
Contribution of solutions	To respond to unmet needs, what solutions do you think that could be carried out?
Other needs	Apart from the personnel, material, economic and emotional/social needs, what other needs of chronic patients have not been mentioned yet?
HCSW training	To what extent do you feel qualified to perform your daily job attending to people in their homes? Do you feel that you may lack training to carry out your daily job attending to people in their homes? To what extent do you feel capable of detecting signs and symptoms that indicate that the people with dependency you attend to in their homes could be getting worse or even showing that their lives could be at risk?

Abbreviation: HCSWs, homecare support workers. Source: developed by author.

**Table 4 ijerph-17-03166-t004:** List of themes and subthemes.

Themes	Subthemes
1. Psychosocial and emotional sphere	1.1. Loneliness as an endemic harm
	1.2. Lack of psychological attention to people with dependency
	1.3. Role of the family (management of bureaucratic issues; knowing the real needs)
	1.4. Fear of hospital admission
2. Legislation and financial issues	2.1. Importance of the Dependency Law
	2.2. Greater expense with greater dependency
3. Possible solutions suggested by HCSWs	3.1. No benefit increase
	3.2. Involvement of HCSWs in the assessments and re-assessments
4. In need of continuous training	4.1. Capacity to recognise versus capacity to act
	4.2. Performing delegated activities
	4.3. Asking for more knowledge

Source: prepared by the authors.

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
