# Peer review of "A Qualitative View of Homecare Support Workers on Unmet Health Needs of People with Dependency"

_ijerph, 2020, doi:10.3390/ijerph17093166_

Round 1
Reviewer 1 Report
I enjoyed your paper very much.
Abstract.
12 “Homecare support workers provides this” should be changed to PROVIDE this
12 rather than say that homecare support workers “directly know” the unmet health needs of those receiving care, I think it would be better to say something like “Homecare support workers may have unique insights into the unmet health needs of those receiving homecare assistance.”
26-27 I think you should change the last part of the second to last sentence to: . . . “although they believe that some delegated activities are beyond their levels of competency,” or something like that.
Crucial: you need one sentence explaining that your study is an analysis of homecare workers in Spain. This is needed in the Abstract, as well as a clear, explicit statement about it very early on in the Introduction.
Introduction
I think you should make reference to the fact that ADLs and IADLs very well may not capture all the complexities in providing care. For example, see Carol Levine “A 21st Century Job Description for Family Caregiving, " Health Affairs Blog, June 14, 2019.DOI: 10.1377/hblog20190612.959252. In addition, there is a reference in this blog to the AARP and Carol Levine report Home Alone Revisited: Family Caregivers Providing Complex Care.” While these references are based upon the US experience, the ideas apply internationally.
The Carol Levine and AARP references I just mentioned concern family caregivers. While your study is of homecare workers, I think would be very important for you to acknowledge the role of family members and friends who help provide care, and the interaction between them and homecare workers.
59 you say “familiar. . . support” – do you mean “familial” as in family members?
Materials and Methods
85 delete the last word of that sentence, “with”
Table 1. All of the participants are female. As is the case in the US and internationally, most homecare workers are female – but 100% female for your participants may open you up for criticism. Did you attempt to recruit males? What percent of homecare workers in Spain are female? I think you should consider adding a couple of sentences addressing this point.
Table 1. Another point – besides gender, can you compare the average age, marital status, education, years of experience and number of people attended for all homecare workers in Spain to your study’s participants?
I addition, would it be possible to know the nationality/ethnicity of HCWs in Spain? In the US, a disproportionately high percentage are Black (African Amerian).
Lastly, it would be helpful to know the average income of HCSWs in Spain. In the US, for example, pay for homecare workers is very low, and job turnover is very high (partly due to low income, partly due to the stress of the job).
Results
124 delete the word “obtained” from the first sentence “The obtained results. . . .”
Very well presented Results. Among other things, I enjoyed the direct quotes and insights of homecare workers.
Discussion
I think you can be a tad clearer here and elsewhere that you not only considered unmet health needs of care recipients, but also of the training and other HCSW-specific issues you analyzed. Indeed, of six themes, four related to unmet needs of care recipients, and two related to HCSWs: “contributions of improvements and the need of these professionals for continuous training.” I think minor revisions in every section making this clear is warranted – and you can (and probably should) mention how better training of HCSWs etc could help reduce unmet health needs of care recipients. You do make this point here and there – and your concluding remarks are exactly about this. I guess I would simply like to see mention of this link between training of HCSWs and quality of care of care recipients earlier, and more explicitly, in the paper (and repeated a couple times)
Reviewer 2 Report
Dear Authors
Thank you for your contribution. Subsequently, some points for improvement are indiciated.
Abstract:
Lines 11: This sentence needs rephrasing. The way it is written, the needs of the homecare support workers are at stake.
This abstract needs to be revised according to the suggestions in the main manuscript below.
Introduction:
The target population requires more detailed description. The definition provided appears to be a translation. If that is the case, please provide some indication that this citation is a translation by the authors. Accurate citations are usually referenced with the respective page. Please add this information.
The target population appears to comprise persons aged over 65 years old, suffering from chronic multimorbidity and having reduced functionality. If that is the case, the ICF (International Classification of Functioning: https://www.who.int/classifications/icf/en/) may need to be referenced. At the least, more details are needed for readers to comprehend the problems regarding this population.
Lines 39/40: The term “activities of daily living” usually references nursing theories such as the one by Virginia Henderson (Henderson & Nite, 1978), Dorothea Orem (Orem et al., 2003) or from Roper et al. (2000). Each one of these theorists have defined needs. It seems important to provide a description and definition of these needs in the target population. Therefore, more literature needs to be provided on the potential problems that the target population may encounter. It can be useful to consult the following articles:
- Cooper, J., & McCarter, K. A. (2014, Jan-Feb). The development of a community and home-based chronic care management program for older adults. Public Health Nurs, 31(1), 36-43. https://doi.org/10.1111/phn.12049
- Garcia-Sanchez, F. J., Martinez-Vizcaino, V., & Rodriguez-Martin, B. (2019, Sep). Conceptualisations on home care for pressure ulcers in Spain: perspectives of patients and their caregivers. Scand J Caring Sci, 33(3), 592-599. https://doi.org/10.1111/scs.12652
- Hale, D., & Marshall, K. (2018, Sep/Oct). Healthy Aging: How It's Done. Home Healthc Now, 36(5), 326. https://doi.org/10.1097/NHH.0000000000000702
- Johnson, Y. O. (2015, Jan). Home care nurses' experiences with and perceptions of elder self-neglect. Home Healthc Now, 33(1), 31-37. https://doi.org/10.1097/NHH.0000000000000169
- Lee, T., & Schiller, J. (2015, Feb). The Future of Home Health project: developing the framework for health care at home. Home Healthc Now, 33(2), 84-87. https://doi.org/10.1097/NHH.0000000000000193
- Seow, H., Barbera, L., Pataky, R., Lawson, B., O'Leary, E., Fassbender, K., McGrail, K., Burge, F., Brouwers, M., & Sutradhar, R. (2016, Feb). Does Increasing Home Care Nursing Reduce Emergency Department Visits at the End of Life? A Population-Based Cohort Study of Cancer Decedents. J Pain Symptom Manage, 51(2), 204-212. https://doi.org/10.1016/j.jpainsymman.2015.10.008
- Sheehan, O. C., Kharrazi, H., Carl, K. J., Leff, B., Wolff, J. L., Roth, D. L., Gabbard, J., & Boyd, C. M. (2018, Jan/Feb). Helping Older Adults Improve Their Medication Experience (HOME) by Addressing Medication Regimen Complexity in Home Healthcare. Home Healthc Now, 36(1), 10-19. https://doi.org/10.1097/NHH.0000000000000632
- Stewart, S., Wiley, J. F., Ball, J., Chan, Y. K., Ahamed, Y., Thompson, D. R., & Carrington, M. J. (2016, May 10). Impact of Nurse-Led, Multidisciplinary Home-Based Intervention on Event-Free Survival Across the Spectrum of Chronic Heart Disease: Composite Analysis of Health Outcomes in 1226 Patients From 3 Randomized Trials. Circulation, 133(19), 1867-1877. https://doi.org/10.1161/CIRCULATIONAHA.116.020730
- Wang, J., Simmons, S. F., Maxwell, C. A., Schlundt, D. G., & Mion, L. C. (2018, Jul-Sep). Home Health Nurses' Perspectives and Care Processes Related to Older Persons with Frailty and Depression: A Mixed Method Pilot Study. J Community Health Nurs, 35(3), 118-136. https://doi.org/10.1080/07370016.2018.1475799
- Wang, M. W., & Huang, Y. Y. (2016, Jun). Evaluating family function in caring for a geriatric group: Family APGAR applied for older adults by home care nurses. Geriatr Gerontol Int, 16(6), 716-721. https://doi.org/10.1111/ggi.12544 (Cooper & McCarter, 2014; Garcia-Sanchez et al., 2019; Hale & Marshall, 2018; Johnson, 2015; Lee & Schiller, 2015; Seow et al., 2016; Sheehan et al., 2018; Stewart et al., 2016; Wang et al., 2018; Wang & Huang, 2016)
The home health support workers do not appear to be formally trained nurses. It is, therefore, necessary to argue the reference to activities of daily life, as these are traditionally areas for nursing. Please provide more details on the home health support workers that are included in this study, particularly their training. Thus, international readers may be better able to understand the study.
Methods
Based on the introduction, the complexity of the topic is difficult to determine. Further argumentation for the qualitative approach is required. In addition, the selection of the research setting is difficult to understand. Why were these two areas included and not any others?
It seems that homecare support workers as well as recipients of home healthcare were interviewed. Please specify these details.
The methods structure is commonly the following:
- Design
- Research environment
- Target population sampling and inclusion/exclusion criteria as well as specific steps insuring participation
- Data collection instrument description, including socio-demographic and other questionnaires that might have been used
- Data analysis procedures including memoing, bracketing, etc.
- Trustworthiness (please refer to: http://www.qualres.org/HomeLinc-3684.html for more details)
- Ethical considerations
Please note that the research approach mentioned here, does not require “saturation” as such. This particular term is part of the research approaches of Grounded Theory and Ethnography.
An argument should be provided for the use of the content analysis.
Any details on the sample such as gender, etc. should be part of the results.
Results
Usually, the results are introduced with a sentence stating that x amount of categories were found or elicited in the analysis.
Each category is then introduced. Please make sure that the content of the category flows normally. In the first category, there is a contradiction: use and non-use of assistant devices. The question that seems to be behind this problem is why? Why were these devices not or wrongly used.
Lines 150/151: This citation does not seem to belong to this category.
The visual schemes provided in each category are difficult to understand. Are these sub-categories? In what way do they play a role? Please only provide schematics to enhance the results.
Lines 162ff: How can the home health support persons know what the clients need? This section needs to be more detailed. Cheering somebody up and providing anti-depressants does not seem to be the same at all.
Lines 178ff: This citation is unclear. What should be expressed by it?
The term “users” seems odd here. Please check with adequate literature to determine the best terminology. In some countries, persons being cared for by home healthcare are clients.
The schematic on lines 200ff seems similar to the one before. What is its purpose?
It seems important to restructure the whole results section so as to better reflect the findings.
Discussion
The need for this study has not been established in the beginning. Hence, stating that no similar studies exist, is problematic. Please see literature below.
As the training of home health support workers is unknown, it is difficult to determine whether they should be able to identify somatic problems in the clients.
Please review the discussion in the face of the above.
Limitations of this study need to be addressed in detail. Please provide very concrete recommendations.
Best wishes,
Your reviewer.
Literature
Cooper, J., & McCarter, K. A. (2014, Jan-Feb). The development of a community and home-based chronic care management program for older adults. Public Health Nurs, 31(1), 36-43. https://doi.org/10.1111/phn.12049
Garcia-Sanchez, F. J., Martinez-Vizcaino, V., & Rodriguez-Martin, B. (2019, Sep). Conceptualisations on home care for pressure ulcers in Spain: perspectives of patients and their caregivers. Scand J Caring Sci, 33(3), 592-599. https://doi.org/10.1111/scs.12652
Hale, D., & Marshall, K. (2018, Sep/Oct). Healthy Aging: How It's Done. Home Healthc Now, 36(5), 326. https://doi.org/10.1097/NHH.0000000000000702
Henderson, V., & Nite, G. (1978). Principles and practice of nursing (6th ed. / [by] Virginia Henderson, Gladys Nite. ed.). Macmillan ; London : Collier Macmillan.
Johnson, Y. O. (2015, Jan). Home care nurses' experiences with and perceptions of elder self-neglect. Home Healthc Now, 33(1), 31-37. https://doi.org/10.1097/NHH.0000000000000169
Lee, T., & Schiller, J. (2015, Feb). The Future of Home Health project: developing the framework for health care at home. Home Healthc Now, 33(2), 84-87. https://doi.org/10.1097/NHH.0000000000000193
Orem, D. E., Renpenning, K. M., & Taylor, S. G. (2003). Self care theory in nursing : selected papers of Dorothea Orem. Springer Pub.
Roper, N., Logan, W. W., & Tierney, A. J. (2000). The Roper-Logan-Tierney model of nursing. Churchill Livingstone.
Seow, H., Barbera, L., Pataky, R., Lawson, B., O'Leary, E., Fassbender, K., McGrail, K., Burge, F., Brouwers, M., & Sutradhar, R. (2016, Feb). Does Increasing Home Care Nursing Reduce Emergency Department Visits at the End of Life? A Population-Based Cohort Study of Cancer Decedents. J Pain Symptom Manage, 51(2), 204-212. https://doi.org/10.1016/j.jpainsymman.2015.10.008
Sheehan, O. C., Kharrazi, H., Carl, K. J., Leff, B., Wolff, J. L., Roth, D. L., Gabbard, J., & Boyd, C. M. (2018, Jan/Feb). Helping Older Adults Improve Their Medication Experience (HOME) by Addressing Medication Regimen Complexity in Home Healthcare. Home Healthc Now, 36(1), 10-19. https://doi.org/10.1097/NHH.0000000000000632
Stewart, S., Wiley, J. F., Ball, J., Chan, Y. K., Ahamed, Y., Thompson, D. R., & Carrington, M. J. (2016, May 10). Impact of Nurse-Led, Multidisciplinary Home-Based Intervention on Event-Free Survival Across the Spectrum of Chronic Heart Disease: Composite Analysis of Health Outcomes in 1226 Patients From 3 Randomized Trials. Circulation, 133(19), 1867-1877. https://doi.org/10.1161/CIRCULATIONAHA.116.020730
Wang, J., Simmons, S. F., Maxwell, C. A., Schlundt, D. G., & Mion, L. C. (2018, Jul-Sep). Home Health Nurses' Perspectives and Care Processes Related to Older Persons with Frailty and Depression: A Mixed Method Pilot Study. J Community Health Nurs, 35(3), 118-136. https://doi.org/10.1080/07370016.2018.1475799
Wang, M. W., & Huang, Y. Y. (2016, Jun). Evaluating family function in caring for a geriatric group: Family APGAR applied for older adults by home care nurses. Geriatr Gerontol Int, 16(6), 716-721. https://doi.org/10.1111/ggi.12544
Round 2
Reviewer 2 Report
Thank you for the careful revision of the manuscript. Although the English language used has been significantly improved, there remain problems with grammar, tense and spelling. Please review the language issues once again.
Abstract: The last sentence seems unrelated to the rest of the abstract. Please modify.
Introduction:
For the definition of dependency, please consult the English version of the recommendation cited (see attached).
Regarding the assessment instruments (Barthel's and Katz): Why do these stand out?
P. 2, lines 55-60: This section seems superfluous. Please remove or state differently and link to lines 85ff. When mentioning such things like "Virginia Henderson", the theory needs to be introduced properly. The readers of the MDPI may or may not be aware of the details of this theory. Hence, a short introduction is recommended.
P. 2, lines 86/86: This sentence does not make sense in comparison to the preceding sentence. Further editing is needed in order to express the train of thoughts exactly.
P. 2, lines 89-93: Although it was required in the previous review to provide some details on the HCSW, the details provided here are far too specific. Please adjust so that readers outside your country may be able to relate.
Please restructure the introduction to provide a central theme. At present, many different topics are addressed at various places in this section. The common thread is not always present.
Methods:
It is stated that a thematic analysis was conducted. The categories found suggest that content analysis was performed instead of thematic analysis. Please consult:
Elo, S., & Kyngas, H. (2008). The qualitative content analysis process. J Adv Nurs, 62(1), 107-115. doi:10.1111/j.1365-2648.2007.04569.x
or
Mayring, P. (2014). Qualitative content analysis. Theoretical foundation, basic procedures and software solution (free download via Social Science Open Access Repository SSOAR, . Retrieved from http://nbn-resolving.de/urn:nbn:de:0168-ssoar-395173 )
or
Vaismoradi, M., Turunen, H., & Bondas, T. (2013). Content analysis and thematic analysis: Implications for conducting a qualitative descriptive study. Nurs Health Sci, 15(3), 398-405. doi:10.1111/nhs.12048
for more details.
Please simplify table 2. There is much repetitive information.
Results:
There are repetitive sections. Please remove all duplications.
P. 7, lines 218-220: Why did the person cared for not use these materials? Why is this a problem for the HCSW? The section on the materials is repetitive and it is unclear why the provided or not provided materials present a problem. It appears that the following section actually is the main theme: resourcefulness. Please reconsider the analysis and restructure your entire results along these lines.
Discussion:
The first sentence does not tie in with the research question stated at the end of the introduction. Please review the entire manuscript to identify the guiding research question and to provide the adequate discussion.
Author Response
The authors really appreciate the reviewer’s interest for the paper and the time spent on carefully reviewing it.
These new modifications can be found in colour red in the text.
